# Quantitative Depth Profiling Using Online-Laser Ablation of Solid Samples in Liquid (LASIL) to Investigate the Oxidation Behavior of Transition Metal Borides

**DOI:** 10.3390/molecules27103221

**Published:** 2022-05-18

**Authors:** Maximilian Weiss, Thomas Glechner, Victor U. Weiss, Helmut Riedl, Andreas Limbeck

**Affiliations:** 1Institute of Chemical Technologies and Analytics, TU Wien, Getreidemarkt 9, 1060 Wien, Austria; victor.weiss@tuwien.ac.at; 2Christian Doppler Laboratory for Surface Engineering of High-Performance Components, TU Wien, Getreidemarkt 9, 1060 Wien, Austria; thomas.glechner@tuwien.ac.at (T.G.); helmut.riedl@tuwien.ac.at (H.R.); 3Institute of Materials Science and Technology, TU Wien, Getreidemarkt 9, 1060 Wien, Austria

**Keywords:** online-LASIL, quantitative depth profiling, boride thin films, high-temperature oxidation

## Abstract

The increased demand for sustainability requires, among others, the development of new materials with enhanced corrosion resistance. Transition metal diborides are exceptional candidates, as they exhibit fascinating mechanical and thermal properties. However, at elevated temperatures and oxidizing atmospheres, their use is limited due to the fact of their inadequate oxidation resistance. Recently, it was found that chromium diboride doped with silicon can overcome this limitation. Further improvement of this protective coating requires detailed knowledge regarding the composition of the forming oxide layer and the change in the composition of the remaining thin film. In this work, an analytical method for the quantitative measurement of depth profiles without using matrix-matched reference materials was developed. Using this approach, based on the recently introduced online-LASIL technique, it was possible to achieve a depth resolution of 240 nm. A further decrease in the ablation rate is possible but demands a more sensitive detection of silicon. Two chromium diboride samples with different Si contents suffering an oxidation treatment were used to demonstrate the capabilities of this technique. The concentration profiles resembled the pathway of the formed oxidation layers as monitored with transmission electron microscopy. The stoichiometry of the oxidation layers differed strongly between the samples, suggesting different processes were taking place. The validity of the LASIL results was cross-checked with several other analytical techniques.

## 1. Introduction

Corrosion can quickly deteriorate the mechanical properties of a material, especially when exposed to elevated temperatures. The resulting shortening of the material’s lifetime and the limitation of its application range can be addressed by deposition of protective coatings. In this context, transition metal diborides are among the most promising candidates that have gained attention in recent years [1,2,3]. The reason for this behaviour is the boron–metal bond allowing for a favorable combination of metal–ceramics properties such as high strength elastic modulus with high thermal and electrical conductivity [4,5]. Additionally, the extremely high melting points (over 3000 °C) of these materials makes them suitable for future high-temperature applications. Such environments are relevant for cutting tools or the aerospace industry [6]. However, in practice, their application is limited through their poor oxidation resistance [7,8]. Pure borides tend to form a B_2_O_3_ layer upon oxidation, which evaporates at elevated temperatures, a process further enhanced by the presence of water vapor by forming the more volatile boric acid, thus limiting the oxidation resistance [9,10,11]. The high-temperature performance can be enhanced by alloying with elements forming highly stable oxides, such as Si and Al, as these are known to yield low parabolic rate constants for the oxidation reaction [8,12]. Silicon is especially of interest, as its oxidation on the surface of borides gives rise to a protective borosilicate glass-like film [13,14].

Transition metal borides can exist over a wide stoichiometric range, and systems with a plethora of transition metals are under investigation. Through their synthesis via sputter deposition, the amount of the alloying elements, such as silicon, can easily be varied over a wide range [2]. To tune the properties of a material, it is of fundamental interest to understand the processes during the oxidation reaction. Hence, a wide range of methods are employed in this field such as thermogravimetry to measure the reaction kinetics, X-ray diffraction for the resulting phase composition, and electron microscopy for its microstructure [15]. Of particular interest is the composition of the oxide layer and the change in the underlying thin film. A method to characterize these needs to give quantitative results and a sufficient depth resolution to separate the layers. There are several frequently employed methods for the analysis of boride thin films: SIMS (secondary ion mass spectrometry) [16], GD-OES/MS (glow discharge optical emission/mass spectrometry) [17,18], XPS (X-ray photoelectron spectrometry) [19], LA-ICP-MS (laser ablation inductively coupled plasma mass spectrometry) [20], and X-ray-based methods, such as XRF (X-ray fluorescence) and SEM-EDX (scanning electron microscopy with energy-dispersive X-ray analysis), with their specific strengths and drawbacks [18,21]. Of those SIMS, GD-OES/MS and LA-ICP-MS are commonly used for depth profiling. A general limitation of direct solid-state analysis, especially for depth profiling, is the availability of suitable reference materials for quantitative analysis. The availability of matrix-matched standards and certified reference materials (CRMs) is limited, especially for novel classes of materials such as borides [22]. A reference material must match the chemical and physical properties of the sample as closely as possible. In the case of thin films undergoing oxidation, the oxide layer is significantly different in properties compared to the boride-based material; therefore, at least two reference materials are needed to precisely determine the stoichiometry throughout the depth profile.

A promising solution to this limitation is the recently introduced online-laser ablation of solids in liquids (online-LASIL) technique [23], as it can circumvent several limitations of other techniques. In online-LASIL, tiny portions of the samples are removed by laser ablation. The ablation is performed in a continuous flow of carrier solution, which transports the removed material into a hyphenated instrument to derive the sample’s composition. The main advantage of online-LASIL is that it is possible to use ready-to-hand liquid standards to determine the composition of the particle suspension produced in the LASIL process. It is well known from slurry analysis and single-particle ICP-MS (spICP-MS) in the ICP plasma, that particles below a specific size behave like ions and can therefore be quantified with liquid standards [24,25]. Through this, the use of matrix-matched standards or certified reference materials (CRMs) can be circumvented. Laser ablation in liquid has been successfully applied to several material science samples in the past [26,27]. The flexible preparation of customized liquid standards allows for quantitative investigations of a wide range of materials not accessible with more conventional techniques. The second main advantage of online-LASIL is the improved depth resolution. Whereas with conventional nanosecond laser ablation, depth resolutions in the order of some hundred nm are possible [18], previously published online-LASIL applications report values below 50 nm [23,28]. With these characteristics, online-LASIL is a promising candidate to perform quantitative, depth-resolved analysis to understand the oxidation behavior of transition metal diboride.

In a preceding work [15], the oxidation behavior of several Si-alloyed transition metal diborides was investigated. As the systems based on chromium diboride doped with silicon showed an exceptionally good oxidation resistance, forming oxide scales with an average thickness of only 400 nm, they are of particular interest. For this work, two samples with different silicon contents were chosen to observe the influence of silicon on oxide scale formation. A methodology based on online-LASIL was developed to perform quantitative depth profile measurements of these materials, enabling the determination of the oxide scale and changes in the stoichiometry of the underlying thin film. Classical acid digestion with subsequent liquid ICP-OES was performed to obtain the bulk stoichiometry before the oxidation. With the data from the rigorous characterization of the thin films, including X-Ray diffraction (XRD), thermogravimetry (TGA), and transmission electron microscopy (TEM), the information of the online-LASIL depth profiles could be validated and put into a material science context.

## 2. Results and Discussion

### 2.1. Optimization of Carrier Solution

From conventional liquid sample analysis, it is well known that analytes, primarily metal ions, can become lost during sample storage or analysis due to the fact of precipitation or adsorption to the walls of the sample container or the analysis system. Therefore, in classical trace elemental analysis, samples and standards are prepared in a diluted acidic solution, usually 1% (*v*/*v*) HNO_3_ or HCl, to stabilize the dissolved ions and prevent possible losses during storage and measurement. When performing laser ablation in liquid, the ablated material can be present in the form of suspended particles but also as dissolved ions. For online-LASIL, this is an especially critical issue as the use of a liquid carrier for the transport of the reaction products from the ablation site to the detection unit might result in fractionation effects preventing an accurate determination of the sample stoichiometry. Possible reasons for the loss of dissolved species have been mentioned above. However, particles can also become lost, for example, through agglomeration and subsequent sedimentation. Thus, to avoid fractionation effects, stabilization of both forms, particles and solvated species, is necessary to prevent losses of these ablation products. For this purpose, the design of the online-LASIL cell has been refined [29] to include two inlets for liquid solutions: one for the carrier solution, which flows over the sample surface, and one for the make-up solution, which is mixed with the carrier solution right after the ablation process. As the flow of the carrier solution was chosen to be higher than the make-up solution, it was ensured that the make-up solution could not be directed into the sample cavity, impeding contact with the sample in the LASIL cell. This concept allows for the use of more concentrated acids to prevent a loss of analytes, as the concentrated acid solution does not come into contact with the sample, which is a precondition for the investigation of acid-sensitive samples.

To find an optimal combination for the composition of carrier and make-up solution, the approach presented in Weiss et al. (2021) [28] was applied as exemplified in Figure 1. A series of standard solutions of the elements of interest were prepared in different candidate make-up solutions. The candidate make-up solution and a candidate carrier solution were flown through the cell until a stable background signal was reached. Then, a defined amount of a spiking material was added to the make-up solution, causing a sudden increase in the respective ICP-MS signals. After the signals in the ICP-MS reached a plateau for all elements, the spike solution was exchanged to the pure make-up solution again. After the signal stabilized again, a washing step with 10% HCl was performed (flown through both inlets). If the analyte was adsorbed onto the walls of the system, in this washing step, a strong peak in the signal would appear; if not, the analyte signal would remain at the baseline level. The lowest acid concentration in the make-up and carrier solutions able to achieve this behavior was considered optimal. Based on the experience reported in [28], for the carrier solution, a NH_4_Cl/HCl buffer with EDTA added was used. A concentration of 910 mmol NH_4_Cl and 30 mmol/L HCl resulting in a pH of 5 and an EDTA concentration of 4.17 mg/L for the make-up solution and a mixture of 2% (*v*/*v*) HCl and 0.2% (*v*/*v*) HF for the make-up solution were found to give optimal results, as HF is known to be necessary to stabilize transition elements [30].

### 2.2. Quantification of LASIL Measurements

In online-LASIL measurements, the material was at least partially ablated in the form of particles. To use liquid standards, it is a prerequisite that these particles behave in the plasma the same way as the liquid standards. This assumption is also routinely used in single-particle ICP-MS (spICP-MS), where liquid standards are used to infer the size of nanoparticles typically below 100 nm, which have gained a broad spectrum of applications over the last years [25,31]. On the upper end, it was found that particles with a diameter of up to 3 µm can be quantified with liquid standards in slurry analysis [24]. To ensure that the particles generated with online-LASIL are small enough to be fully ionized in the plasma and fulfill the requirements for the application of liquid standard solutions, measurement of the size distribution was necessary. For this purpose, the flow out of the online-LASIL cell was collected in a tube during the ablation. The solution containing the generated nanoparticles was analyzed in a ZetaView particle tracker after dilution with water to provide a signal in the working range of the instrument. Particles in the range between 65 and 200 nm were found with a median of 114 nm, which is the size range typically studied in spICP-MS and one order of magnitude below the size limit reported for slurry samples. In Appendix A, the size distribution graph is shown. Thus, the use of liquid standards is a valid approach for quantification of the particle suspensions produced with online-LASIL.

Online-LASIL measurements were quantified using the standard addition approach [32] by adding defined amounts of the investigated analytes to the make-up solution. In Figure 2, a typical time-resolved signal for ^52^Cr of a depth profile measurement is demonstrated.

The concentration of the elements in the ablated material can be calculated by the following formula:(1)$$c_{abl}=\frac{c_{s}}{S}\left(A-S\right)\;$$
where $c_{abl}$ is the concentration of the respective element in the flow from ablation; $c_{S}$ is the concentration of the spike; *A* is the integrated area of the ablation peak; *S* is the integrated area from the spike. Note that the integration regions *A* and *S* have the same duration.

To exclude that material is removed from the fused silica window of the LASIL cell during ablation, which would contribute to the silicon signal detected with ICP-MS, and measurements with pure Al_2_O_3_ substrates were performed. In these experiments, no increase in the silicon signal was observed during sample ablation, indicating that the use of a silicon window did not influence the analysis of Si.

### 2.3. Optimization of Depth Profile Measurements

Measurement of silicon with ICP-MS is challenging due to the high ionization energy of 8.151 eV. Moreover, the main isotopes ^28^Si and ^29^Si are interfered with by several polyatomic ions arising from the ICP plasma [33]. Therefore, the detection limit for silicon with ICP-MS is orders of magnitude worse than for most other elements [34]. With the advent of collision-reaction cell technology, the intensities of interfering polyatomic ions could be significantly reduced; thus, the signal ratio of analyte to the background can be improved [35]. However, the sensitivity achieved for silicon is still lower compared to the other elements examined in this study. With the use of liquid calibration standards which were introduced via the make-up solution, detection limits (LODs) for the applied ICP-MS procedure were determined. Silicon had, as expected, the highest LOD of 69 ng/g, compared to 2 ng/g for B and 0.08 ng/g for Cr, and is, therefore, the limiting factor in the analysis of the samples.

In the used online-LASIL setup, the main parameter influencing the amount of sample material introduced into the detection unit was the ablation rate, which is determined by the applied laser energy [36,37]. The ablation energy cannot be arbitrarily reduced, as the ablation process only onsets over a certain energy threshold [38]. For optimization a defined laser pattern was ablated using different laser energies, and the peak area of the derived transient signals was integrated. The lowest investigated laser energy to achieve a measurable ^28^Si signal was 0.17 mJ (Figure 3). At this energy, the boride film could be fully ablated with ten layers, indicated by a sudden drop in the total signal observed for the measured elements when the Al_2_O_3_ substrate was reached. With further ablation passes at this energy, it was impossible to observe an ^27^Al signal, indicating that the energy was below the ablation threshold for Al_2_O_3_. These findings were confirmed by a profilometric scan of the ablation crater, which suggests that, in total, approximately 2.4 µm of the sample had been ablated, resulting in a depth resolution of ~240 nm per layer at this laser energy. For experiments with a reduced laser energy of 0.068 mJ, an ablation rate of approximately 85 nm was observed. A further reduction in the laser energy to 0.034 mJ resulted in a decreased ablation rate of 54 nm per layer. However, with these conditions the amount of Si introduced into the ICP-MS was insufficient for quantitative measurements.

This outcome indicates that the sensitivity of the silicon analysis was the limiting factor, as the amount of ablated material decreased concomitantly with the reduction in the thickness of an ablation layer. Nevertheless, for elements with an enhanced sensitivity in ICP-MS analysis, investigations with a better depth resolution were possible. In the case of B, even for the lowest laser energy, excellent signals were obtained (for details see Figure 3), enabling measurements with an ablation rate of 54 nm only. This is in accordance with previous works [23,28], where a similar depth resolution could be achieved.

On the upper bound, the laser energy was limited by the mechanical strength of the fused silica window, which can burst through the generated cavitation bubble in the carrier solution at approximately 1 mJ laser energy. Further, it was observed that the boride thin films delaminated from the substrate if the laser energy was too high.

Other factors influencing the ablation were the spot size and the number of shots per sample location. To reduce cratering effects due to the Gaussian profile of the laser beam, overlapping spots were chosen where the laser hit each ablation spot two times by a stage-velocity of 0.5 mm/s and a laser frequency of 10 Hz.

### 2.4. Measurement of Oxidized Samples

Two silicon-alloyed chromium diboride samples, as stated before, designated as A and B, with different silicon doping levels were investigated within this study and used for the depth profile measurements. Of the as-deposited samples, the bulk stoichiometry was determined with liquid ICP-OES; the values are listed in Table 1, and the standard deviation was derived from three replicates of the digestion. The obtained bulk stoichiometry agreed with the composition expected from the production of thin film.

In the first step, to validate that online-LASIL yields accurate values for the stoichiometry of the samples, measurements with an enhanced laser energy of 0.51 mJ were performed. This high laser energy was not suitable for the depth profile analysis, but the increased ablation rate resulted in higher analyte concentrations in the carrier solution and, thus, improved signal-to-noise ratios for subsequent ICP-MS analysis. As can be seen in Table 1 the findings derived for the native, non-oxidized samples were in good agreement with the values obtained from ICP-OES measurements, in particular when considering that in the case of online-LASIL only tiny sample areas of approximately 0.1 mm^2^ were used for analysis, whereas for the ICP-OES measurements, 5 × 5 mm large pieces were used. The presented averages and standard deviations were determined from three ablation passes at different sample positions. This outcome demonstrates the applicability of the proposed online-LASIL procedure, but it also confirms that the optimized composition of the carrier solution prevented the fractionation effects of the analyte.

In Figure 4, the quantitative online-LASIL depth profiles of the two oxidized samples with BF-TEM cross-sections of the same samples are shown. In the TEM cross-section of sample A, two layers separated by a sharp interface can be seen; the upper corresponded to the formed oxide scale with a thickness of approximately 500 nm, and the lower was the remaining boride film.

The performed online-LASIL measurement revealed a very similar outcome, showing for the first ~500 nm a distinctly different composition when compared to the rest of the sample. The first ablation layer exhibited a stoichiometry of Cr_0.04_Si_0.62_B_0.34_, indicating an enrichment of silicon in the oxide thin film. With the fourth ablation layer, the sample nearly reached its native bulk composition.

The TEM image of sample B shows a rough interface between the oxide scale and the base boride coating, with a thickness varying between 180 and 750 nm with 400 nm in mean. Further investigations [15] indicate that the oxide film consists of an outer Cr_2_O_3_ layer and an inner silicon-rich layer. The irregular interface and the large grains stem from recrystallization processes during the heat treatment and are influenced through the higher temperature and the higher silicon content compared to sample A. X-ray diffractograms [15] obtained from the sample show that Cr_2_O_3_ and Si were present as phases alongside the base material. This was confirmed through the online-LASIL measurements. The first two ablation layers were highly enriched in chromium, showing no boron signal. At a depth of approximately 500 nm, the boron signal started to in increase and reached its bulk value at approximately 1500 nm. Silicon was enriched in the top layers, which agrees with the large grains of silicon visible in the TEM, and reached the nominal sample concentration at a depth of 1500 nm. Interestingly, the composition of the third ablation layer neither matched with the top layer nor with the underlying layer and seemed to be a mixture of the formed oxide and the bulk material, which agrees with the coarse interface visible in the TEM.

The results derived from online-LASIL measurements indicate that no significant change in the overall stoichiometry of the whole film occurred during the oxidation.

## 3. Material and Methods

### 3.1. Reagents and Instrumentation

High-purity (18.2 MΩ resistivity at 25 °C) water was obtained from a Barnstead EASYPURE II system (Thermo Fisher Scientific, Waltham, MA, USA). Acids and all other chemicals not otherwise mentioned and certified ICP liquid standards were purchased from Merck (Darmstadt, Germany) in at least analytical quality. Laser ablation was performed with a J200 Tandem (Applied Spectra Inc., Sacramento, CA, USA) equipped with a 266 nm Nd:YAG laser with a pulse duration of 5 ns. For the ICP-MS measurements, an iCAP Qc, (Thermo Fisher Scientific, Bremen, Germany) equipped with an HF (hydrofluoric acid)-resistant sample introduction kit (alumina injector tube, a perfluoroalkoxy alkane (PFA) cyclonic spray chamber, and a PFA concentric nebulizer) was used. Liquid ICP-OES measurements were performed on an iCAP 6500 RAD (Thermo Fisher Scientific, USA) coupled to an ASX-520 autosampler (CETAC Technologies, Omaha, NE, USA) equipped with an HF-resistant sample introduction kit, which included a Miramist nebulizer (Burgener Research, Mississauga, ON, Canada), a Teflon spray chamber, and an Al_2_O_3_ injector tube. TEM measurements were performed on an FEI TECNAI F20 (Thermo Fisher Scientific, Bremen, Germany). The depth of ablation craters was recorded on a Dektak XT (Bruker, Billerica, MA, USA) stylus profilometer. Particle size distribution of the ablated material was conducted on a ZetaView particle tracker (Particle Metrix, Inning am Ammersee, Germany). Data analysis was performed in Excel (Microsoft Cooperation, Redmond, MA, USA) and OriginPro 2020 (OriginLab Corporation, Northampton, MA, USA).

### 3.2. Deposition and Treatment of Samples

The deposition of the coatings, the oxidation treatment, and rigorous characterization of the samples are described in detail by Glechner et al. [15]. The two samples, designated as A and B in the following, were deposited in an in-house built magnetron sputter device [39] using a CrB_2_ target (Plansee Composite Materials GmbH, Lechbruck am See, Germany) on which Si wafer plates (CrysTec GmbH, Berlin, Germany) were placed on the racetrack to alloy Si. Through changing the number of silicon plates, the amount of silicon in the samples varied. Polycrystalline Al_2_O_3_ (CrysTec GmbH) platelets were used as substrate. The thickness of the deposited films was approximately 2.4 µm. Oxidation treatment of the samples was performed in a chamber box furnace under ambient air. The temperatures used were 1100 °C for sample A or 1200 °C for sample B for 3 h.

### 3.3. LASIL Setup

For the online-LASIL measurements, specimens were put in a liquid-tight in-house made cell, which formed the flow path. The cell design was similar to previous works [28,29,32]. The LASIL cell (Figure 5) consisted of a PEEK (polyether ether ketone) body with a 5 × 5 × 0.5 mm pocket to contain the sample. The employed LASIL cell design contained two inlets and one outlet for fluids. The cell was sealed by a PDMS (polydimethylsiloxane) film (500 µm thick) containing the 500 µm wide flow path to guide the liquid over the sample. A fused silica window, transparent to the UV laser wavelength, was placed on top and aligned to the body by a counterpart made of PEEK. A peristaltic pump (Perimax 12, SPETEC, Erding, Germany) was used to transport the required fluid flows through the system. All tubing was made out of PFA with an inner diameter of 0.5 mm on the input side and 0.25 mm between the LASIL cell and the ICP-MS instrument.

The LASIL cell was positioned on the movable XYZ sample stage of the J200 laser ablation platform. To analyze the ablated material, the generated particle suspension was purged with a carrier solution into a quadruple ICP-MS. The instrument was tuned daily for a maximum ^115^In signal intensity and a minimum ^140^Ce^16^O/^140^Ce oxide ratio. The ICP-MS instrument was operated in the KED (kinetic energy discrimination) mode using 7% H_2_ in He as collision gas. The ICP-MS measurement parameters are listed in Table 2. The instrument software (Qtegra version 2.10) was used for data collection and evaluation.

For LASIL measurements, substrates were broken into 5 × 5 mm pieces by scratching with a diamond cutter to fit tightly into the pocket of the LASIL cell. The current LASIL setup allowed for the use of two fluid flows: a carrier flow and a make-up flow (for details, see Figure 5); both contained 10 ng/g of indium as an internal standard to monitor potential signal drifts during measurements. The isotopes selected for ICP-MS analysis, as well as the applied instrumental parameters, are compiled in Table 2. Laser ablation was performed in the line scan mode, and the parameters are stated in Table 3.

### 3.4. ICP-OES Reference Measurements

To obtain reference values for the samples, an aliquot of the native, unoxidized samples was converted into a solution and measured with conventional liquid ICP-OES. For this purpose, samples were broken into pieces of approximately 5 × 5 mm and digested in triplicates in metal-free falcon tubes with a mixture of 0.25 mL nitric acid and 0.25 mL hydrofluoric acid at a temperature of 80 °C for 10 min. Derived solutions were diluted to a final volume of 20 mL with ultrapure water, and europium was added as an internal standard with a final concentration of 1 µg/g. External calibration with matrix-adjusted standards was used for quantification. Two emission lines were observed per element, one used for quantification one for quality control; for further details, see Appendix A. The applicability of this procedure was recently demonstrated [6,20,29].

## 4. Conclusions

In this work, the oxidation behavior of two samples of chromium diboride doped with different levels of silicon was investigated. TEM images revealed a clear difference in the oxidation resistance of the two coating materials. Online-LASIL has been applied for the determination of sample stoichiometry but also for the measurement of quantitative depth profiles. Further improvement of the cell design and careful optimization of the measurement conditions enabled, for the first-time, quantitative measurements of depth profiles without the use of matrix-matched standards.

The results obtained for the native, unoxidized samples were found to be in good agreement with ICP-OES reference measurements, demonstrating the suitability of the proposed online-LASIL approach for the analysis of Si-alloyed transition metal diborides. The derived depth profiles showed a good correlation with TEM images of the samples. Moreover, with online-LASIL, it was possible to gain insight into the exact stoichiometry of the oxide layer and the change in the bulk sample below. It could be shown that the oxide layers of the two samples had a very different composition. This information is not easily accessible with other techniques capable of depth-resolved measurements, as matrix-matched standards are required for most methods. In this case, where the sample consisted of two different materials (i.e., the oxide layer and the native boride), at least two CRMs would be needed for quantitative investigations. This is a difficult task since, for many materials, it remains challenging to find even one suitable CRM.

As the capabilities of online-LASIL for the assessment of depth-resolved changes in thin film stoichiometry have been shown and validated by several reference techniques, it is intended to extend the investigations to a larger number of samples in the future. These should cover a broader range of sample compositions and treatment conditions, providing more insights into the fundamental processes of high-temperature corrosion.

## Figures and Tables

**Figure 1 molecules-27-03221-f001:**
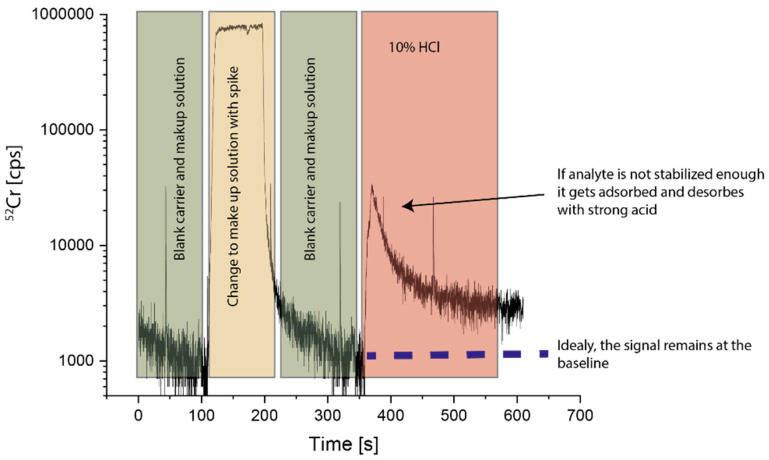
Demonstration of the optimization process for the make-up and carrier solutions: If the analyte was not stabilized enough, it was absorbed onto the walls, and it appeared as a peak if it was desorbed by concentrated acid.

**Figure 2 molecules-27-03221-f002:**
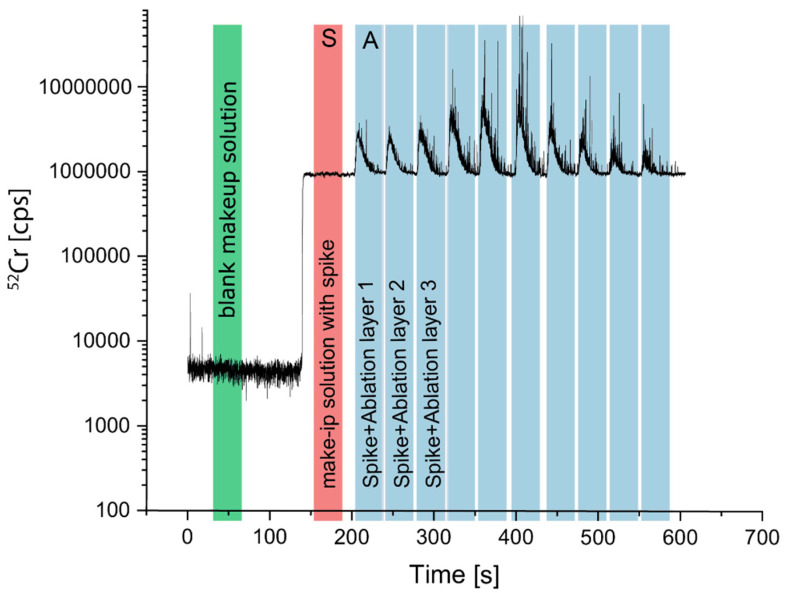
Illustration of the signal during a depth profile measurement with standard addition.

**Figure 3 molecules-27-03221-f003:**
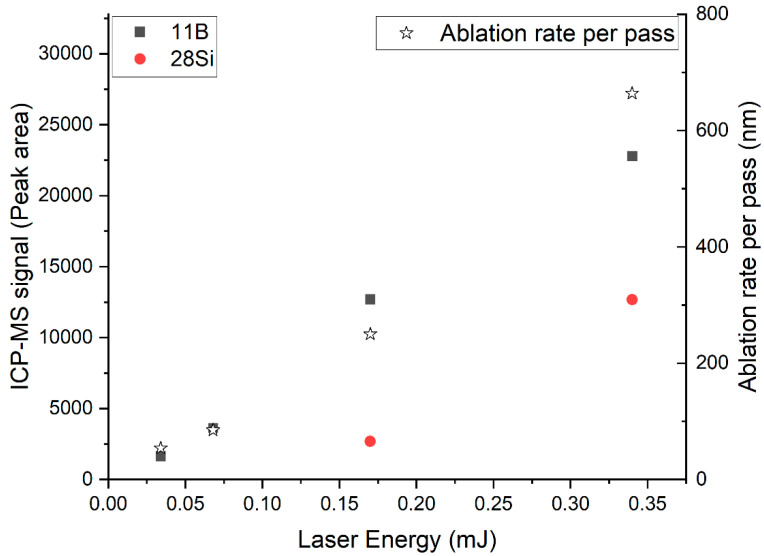
Dependence of the ablation rate and the ICP-MS signal from the applied laser energy.

**Figure 4 molecules-27-03221-f004:**
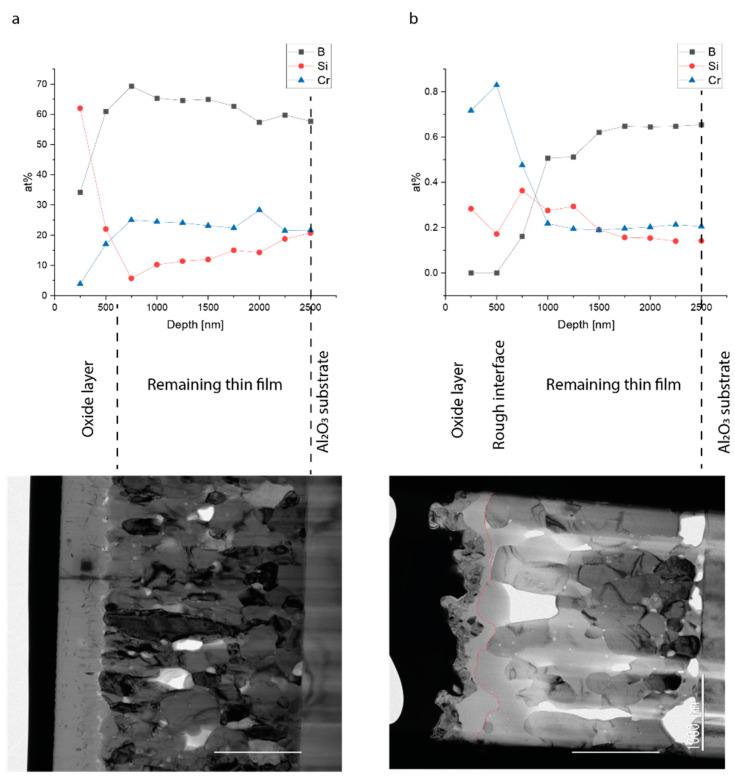
LASIL depth profiles with corresponding bright-field TEM cross-sections of the samples: (**a**) sample A (native stoichiometry Cr_0.27_Si_0.9_B_0.64_); (**b**) sample B (native stoichiometry Cr_0.26_Si_0.16_B_0.58_). The scale bar represents 1000 nm, and the images were rotated to correspond to the depth profiles. The thin red line in the TEM image of sample B indicates the boundary of the oxide and was determined with high-angle annular dark-field imaging (HAAFD) STEM (scanning transition electron microscopy).

**Figure 5 molecules-27-03221-f005:**
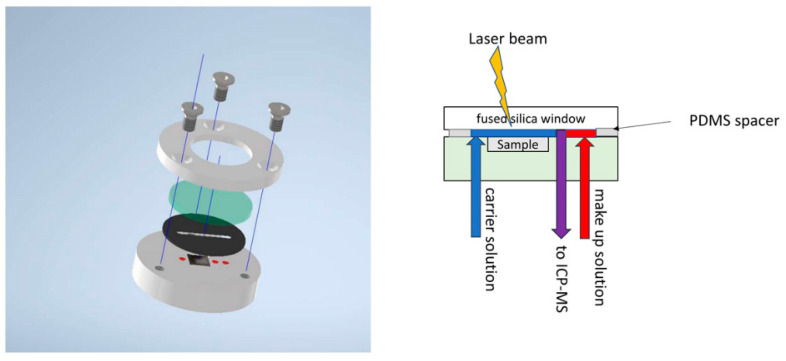
(**Left**): CAD exploded drawing of the LASIL cell. The gray parts represent PEEK, the PFA tubing pushed in the PEEK body is depicted in red, the PDMS spacer in black, and the fused silica window in green. (**Right**): Cross-section of the LASIL cell showing the fluid path and the flows through the sample cell. The LASIL cells consisted of a PEEK body with a cavity for the sample. Through the body, three tubes pushed the flow for two inlets and one outlet. The carrier flowed over the sample, the make-up solution was added behind the sample in the fluid path, so it did not come into contact with the sample. The two flows mixed just before they left the LASIL cell at the outlet to the ICP-MS instrument. The cell was sealed by a PDMS spacer and a fused silica window.

**Table 1 molecules-27-03221-t001:** Comparison of the results of the stoichiometry determination between the liquid ICP-OES (*n* = 3) and the online-LASIL analysis performed on the native samples with high laser energy (0.51 mJ). The cations were normalized to 100%.

Sample	Measurement	Cr at%	Si at%	B at%
Sample A	ICP-OES	27.1 ± 0.1	8.9 ± 0.3	64.0 ± 0.3
	LASIL bulk analysis	25.0 ± 1.8	7.7 ± 3.2	67.3 ± 1.5
Sample B	ICP-OES	25.6 ± 0.2	15.8 ± 0.1	58.6 ± 0.1
	LASIL bulk analysis	22.8 ± 1.0	18.8 ± 1.5	58.4 ± 0.6

**Table 2 molecules-27-03221-t002:** ICP-MS measurement parameters for online-LASIL.

Parameter	Value
RF power	1550 W
Auxiliary gas flow (Ar)	1.0 L/min
Cooling gas flow (Ar)	14 L/min
Nebulizer gas flow (Ar)	0.8 L/min
CCT bias	−21 V
Pole bias	−18 V
KED gas flow (7% H_2_ in He)	5 mL/min
Monitored ions	^11^B, ^27^Al, ^28^Si, ^52^Cr, ^53^Cr, ^115^In,
Dwell time	0.01 s for ^27^Al, ^52^Cr, ^53^Cr, ^115^In 0.05 s for ^11^B and ^28^Si

**Table 3 molecules-27-03221-t003:** Laser parameters for the online-LASIL measurements.

Parameter	Value
Laser energy depth profile	0.17 mJ
Laser energy survey run	0.51 mJ
Spot size	100 µm
Scan speed	500 µm/s
Carrier solution flow	0.53 mL/min
Makeup solution flow	0.28 mL/min
Repetition rate	10 Hz
Investigated sample area	0.1 mm^2^

## Data Availability

The data presented in this study are available upon request from the corresponding author.

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
