# Peer review of "Quantitative Depth Profiling Using Online-Laser Ablation of Solid Samples in Liquid (LASIL) to Investigate the Oxidation Behavior of Transition Metal Borides"

_molecules, 2022, doi:10.3390/molecules27103221_

Round 1

Reviewer 1 Report

In the manuscript "Quantitative depth profiling using online-laser ablation of solid samples in liquid (LASIL) to investigate oxidation behavior of transition metal borides" by Weiss et al., a new method to investigate the oxidation resistance of  new materials has been developped and applied to the investigation of  the forming oxide layer and the change in the composition of the remaining thin film of chromium-diboride doped with silicon. The article is interesting and the obtained results are useful.The topic is original and relevant to the field and the new method is an extension giving new possibilities for investigations of oxidation resistance of new materials.   It is well organized and methodolgy is well explained. The manuscript is clearly written in good English. The conclusions are consistent with the findings and supported by the obtained results. The references are appropriate  and cover well the investigated topic. I recommend its publication in present form.

Misprints:

line 125: The design --> the design
line 260: perfumed --> performed 

Author Response

Answer: We thank the reviewer for the encouraging review. The misprints have been corrected in the manuscript.

Reviewer 2 Report

The presented manuscript describes an interesting analytical approach of quantitative depth profiling of layered samples using liquid standards and a special ablation cell for LA-ICP-MS. There is no need for matrix-matched standards for LA. The method looks very promising on the first look. However, it is difficult to understand the text. It should be revised thoroughly before possible publication.

The current status is not properly separated from the results part. Many things are referenced to previous publications. The authors should clearly declare what is the novelty of this work against the already published results.

It is also questionable if two depth profiles are sufficient for a fullpaper. It might be rather published as a technical note provided that the authors do not add more experimental results.

Spoken about an optimization I would expect some graphs signal vs any parameters (laser energy, spot size, MS dwell time,...). I believe that the authors adjusted the appropriate instrumental conditions but this process is not what I understand under the optimization.

Do I understand correctly that the sample was scanned with the laser beam 100 um in diameter on the surface several times in the same line? If the scan speed is 1 mm/s the craters are not overlapped at the repetition rate of 10 Hz. The scan line was 1 mm long at the ablated surface area 0.1 mm2 so that the ablation time per line is 1 s? Page 6, lines 233-236 “To reduce cratering effects due to the Gaussian profile of the laser beam, overlapping spots were chosen where the Laser hits each ablation spot two times by a stage-velocity of 1 mm/s and a laser frequency of 10 Hz.“ makes me no sense. Any overlapping line of craters deepens the ablation groove and the measured signal represents another depth. How can the crater effect be reduced in this way, please?

Table 1 – the sample composition was evidentially normalized to 100 % from 3 elements. This is not mentioned. Can you also show the rough concentrations to verify the quantification approach? The formula (1) does not include any corrections (matrix effects, internal standard). What about possible oxides in the sample (page 8)?

I would welcome a microscopic picture of the lateral cut with the ablation path. Is the ablation rate the same in the various parts of the depth profile?

Fig. 4 – the drawing is nice but it deserves better description. The text info is difficult to link with the drawn parts. I wonder that the silica window is not ablated because of the very short distance between the sample and the window.

Reviewer 3 Report

In this work, the author reported an analytical method for the quantitative measurement of depth profiles without using ma-trix-matched reference materials. By using this approach and based on the recently introduced online-LASIL technique, thay can achieve a depth-resolution of 240 nm.  Two chromium-diboride samples with different Si contents suffering an oxidation treatment were used to demonstrate the capabilities of this technique. The work is interesting and worth for publication. The paper is weill written and can be accepted without revision.

Author Response

We thank the reviewer for the encouraging review.

Round 2

Reviewer 2 Report

The authors addressed the reviewer's comments and answered the questions. The work can now be published in my opinion.